# Population Pharmacokinetics of Amikacin in Patients on Veno-Arterial Extracorporeal Membrane Oxygenation

**DOI:** 10.3390/pharmaceutics14020289

**Published:** 2022-01-26

**Authors:** Claire Pressiat, Agathe Kudela, Quentin De Roux, Nihel Khoudour, Claire Alessandri, Hakim Haouache, Dominique Vodovar, Paul-Louis Woerther, Alice Hutin, Bijan Ghaleh, Anne Hulin, Nicolas Mongardon

**Affiliations:** 1Laboratoire de Pharmacologie, DMU Biologie-Pathologie, Assistance Publique des Hôpitaux de Paris (APHP), Hôpitaux Universitaires Henri Mondor, 94010 Créteil, France; claire.pressiat@aphp.fr (C.P.); Nihel.Khoudour@aphp.fr (N.K.); bijan.ghaleh@inserm.fr (B.G.); anne.hulin@aphp.fr (A.H.); 2Faculté de Santé, Université Paris Est Créteil, 94010 Créteil, France; agathe.kudela@aphp.fr (A.K.); quentin.deroux@aphp.fr (Q.D.R.); paul-louis.woerther@aphp.fr (P.-L.W.); 3Inserm U955-IMRB, Equipe 03 “Pharmacologie et Technologies pour les Maladies Cardiovasculaires (PROTECT)”, Ecole Nationale Vétérinaire d’Alfort (EnVA), Université Paris Est Créteil (UPEC), 94700 Maisons-Alfort, France; alice.hutin@aphp.fr; 4Service d’Anesthésie-Réanimation Chirurgicale, DMU CARE, DHU A-TVB, Assistance Publique-Hôpitaux de Paris (AP-HP), Hôpitaux Universitaires Henri Mondor, 94010 Créteil, France; claire.alessandri@aphp.fr (C.A.); hakim.haouache@aphp.fr (H.H.); 5INSERM UMRS 1144, 75004 Paris, France; dominique.vodovar@aphp.fr; 6Université de Paris, 75010 Paris, France; 7Université Paris-Saclay—CEA—CNRS—Inserm—BioMaps, 91400 Orsay, France; 8Centre Anti-Poison, Fernand Widal Hospital, AP-HP, University of Paris, 75010 Paris, France; 9Service de Réanimation Médicale et Toxicologique, Hôpital Lariboisière, Assistance Publique des Hôpitaux de Paris, (APHP), 75010 Paris, France; 10Département de Prévention, Diagnostic et Traitement des Infections, Hôpitaux Universitaires Henri Mondor-Albert Chenevier, Assistance Publique-Hôpitaux de Paris (AP-HP), 94010 Créteil, France; 11Research Group Dynamic, Faculté de Santé de Créteil, Université Paris-Est Créteil Val de Marne (UPEC), 94010 Créteil, France; 12SAMU de Paris, Hôpital Universitaire Necker Enfants Malades, Assistance Publique-Hôpitaux de Paris, 75015 Paris, France

**Keywords:** extracorporeal membrane oxygenation, sepsis, amikacin, pharmacokinetic/pharmacodynamic modeling, population pharmacokinetics, Bayesian modelization, therapeutic drug monitoring

## Abstract

Veno-arterial extracorporeal membrane oxygenation (V-A ECMO) support leads to complex pharmacokinetic alterations, whereas adequate drug dosing is paramount for efficacy and absence of toxicity in critically ill patients. Amikacin is a major antibiotic used in nosocomial sepsis, especially for these patients. We aimed to describe amikacin pharmacokinetics on V-A ECMO support and to determine relevant variables to improve its dosing. All critically ill patients requiring empirical antimicrobial therapy, including amikacin for nosocomial sepsis supported or not by V-A ECMO, were included in a prospective population pharmacokinetic study. This population pharmacokinetic analysis was built with a dedicated software, and Monte Carlo simulations were performed to identify doses achieving therapeutic plasma concentrations. Thirty-nine patients were included (control *n* = 15, V-A ECMO *n* = 24); 215 plasma assays were performed and used for the modeling process. Patients received 29 (24–33) and 32 (30–35) mg/kg of amikacin in control and ECMO groups, respectively. Data were best described by a two-compartment model with first-order elimination. Inter-individual variabilities were observed on clearance, central compartment volume (V_1_)_,_ and peripherical compartment volume (V_2_). Three significant covariates explained these variabilities: Kidney Disease Improving Global Outcomes (KDIGO) stage on amikacin clearance, total body weight on V_1,_ and ECMO support on V_2_. Our simulations showed that the adequate dosage of amikacin was 40 mg/kg in KDIGO stage 0 patients, while 25 mg/kg in KDIGO stage 3 patients was relevant. V-A ECMO support had only a secondary impact on amikacin pharmacokinetics, as compared to acute kidney injury.

## 1. Introduction

Veno-arterial ExtraCorporeal Membrane Oxygenation (V-A ECMO) support is increasingly used for critically ill patients with refractory cardiogenic shock [1]. These patients are highly exposed to infectious complications [2,3]. Indeed, more than half of these patients will require antimicrobial therapy during their support, with inherent increased morbidity and mortality [4,5,6,7].

Drugs pharmacokinetics, notably antibiotics, can change in critically ill patients undergoing ECMO, which increases the apparent volume of distribution [8] and alters drugs clearance [9]. This may lead to either therapeutic failure or drug toxicity.

Because of the high prevalence of gram-negative bacteria in ICU patients, amikacin is often used in combination with beta-lactams for empirical antimicrobial therapy. In adults with normal renal function, 94–98% of a single IM or IV dose of amikacin is excreted unchanged by glomerular filtration in the kidney within 24 h. However, data on amikacin pharmacokinetics in critically ill patients on V-A ECMO are limited and often outdated, using inadequate therapeutic dosing [10,11,12,13]. While drug clearance is highly variable in critically ill patients, only one study has considered the increase of amikacin clearance in patients undergoing ECMO [14]. Few studies described amikacin therapeutic drug monitoring in patients supported by ECMO: maximal concentrations were often below the recommended concentrations with the standard dosing [10,11].

The aims of this study were to describe amikacin pharmacokinetics in a population of critically ill patients treated or not with V-A ECMO using an original population pharmacokinetics approach, to determine the variables explaining its variability and to simulate optimal amikacin dosing.

## 2. Patients and Methods

### 2.1. Patients and Setting

This study was conducted in the surgical intensive care unit at Henri Mondor Hospital (Créteil, France) from July 2013 to September 2015. We prospectively included patients requiring amikacin for sepsis. In case of repeated amikacin administration, only the first administration was studied. Exclusion criteria were patient refusal or his next of kin, and any patient presenting one of the following characteristics: age < 18, pregnancy, patients on chronic dialysis or requiring renal replacement therapy during the study period, or documented cirrhosis. Demographic data, reason for admission, Sepsis-related Organ Failure Assessment (SOFA score), Simplified Acute Physiology Score (SAPS II), length of hospitalization, discharge status (alive/deceased) and cause of death were collected for all patients. Data related to sepsis, such as severity of sepsis and bacteriological documentation were also recorded. Organ failure was assessed as follows: Glasgow coma scale for neurological failure; doses of vasopressors (µg/kg/min) and lactatemia (mmol/L) for hemodynamic failure; invasive mechanical ventilation, PaO_2_ and PaCO_2_ (mmHg) for respiratory failure; albuminemia (g/L), prothombine time (PT), V factor, Alanine-Amino-Transferase (AST), Aspartate-Amino-Transferase (ALT), Gamma-Glutamyl-Transferase (GGT) and ALP (IU/L), bilirubinemia (µmol/L) for liver failure; pH, arterial lactate for metabolic failure and 24-h urine output following the injection (mL), uremia (mmol/L) and serum creatinine (µmol/L), and Kidney Disease Improving Global Outcomes (KDIGO) for renal failure. The KDIGO stages acute kidney injury (AKI) into three stages based on serum creatinine and urine output [15,16]. The coadministration of colloids/crystalloids/red blood cell packs was collected, allowing to measure the 24 h input/output balance as hemodilution parameters. Finally, in patients undergoing ECMO, the pump flow rate was recorded.

This study was approved by the ethics committee, *Comité d’Ethique pour la Recherche en Anesthésie-Réanimation* (CERAR, IRB 00010254-2015-027). Written consent was obtained from the patients or their next of kin.

### 2.2. Study Design

Amikacin was administered according to a standardized protocol: 30 mg/kg TBW (total body weight: weight of the day), diluted in 50 mL of 5% glucose solution and continuously infused over 30 min. Doses were rounded up to a multiple of 125 mg.

Blood samples were performed on heparin tubes at the end of the 30 min-infusion, between 3 and 6 h, 7 and 9 h, 9 and 12 h, and 24 h after the amikacin injection. If the 24 h concentration was greater than 2.5 mg/L, another sample was scheduled every 12 h.

### 2.3. Assay

Plasma concentrations of amikacin was assayed by enzyme-linked immunoturbidimetry on an Architect^®^ 4000 (Abbott, Rungis, France). The lower limit of quantification was 0.8 mg/L and the upper limit of quantification 40 mg/L. The method was validated and certified by Health and Quality authorities Cofrac (French accreditation committee). External quality controls were regularly assessed, in accordance with international recommendations.

### 2.4. Therapeutic Amikacin Monitoring

Therapeutic amikacin monitoring in critically ill patients is based on French recommendations [17]. Aminoglycoside monitoring requires to measure the plasma concentrations: maximal concentration (Cmax) for efficacy and minimal concentration (Cmin) for toxicity.

### 2.5. Statistical Analysis

Categorical variables were expressed as number (percentage) and compared using the chi-square test. Continuous variables were expressed as medians (1st; 3rd quartile) and compared using Student′s t test.

### 2.6. Population Pharmacokinetics

Amikacin data were analyzed using non-linear mixed- effect modeling software (MONOLIX^®^, version 2020R1, Lixoft, Antony, France), together with the stochastic approximation expectation–maximization (SAEM) algorithms. Monolix (Non-linear mixed-effects models or “MOdèles NOn LInéaires à effets miXtes” in French) is a platform of reference for model based drug development, used by academia, the pharmaceutical industry as well as the US regulatory agencies. Models were coded with differential equations in a MLXTRAN script file. Various structural models were tested, including one- or two-compartment distribution with first-order absorption and elimination rate constants. Categorial covariates were tested as follows:*θ*_i_ = *θ*_pop_ × *θ^COV^*(1)
where *θ*_i_ is the individual parameter for the patient, *θ*_pop_ is the typical value of the parameter, *θ*^COV^ is the covariate parameter, and COV is the category 0 or 1 for the covariate under study.

Continuous covariates were associated with pharmacokinetics parameters by a power function:(2)$$\theta_{i}=\theta_{pop}\times {\left(\frac{Cov_{i}}{\left(Median\left(Cov\right)\right)}\right)}^{PWR}$$

Other covariates were also tested: demographic characteristics (age, sex, total body weight), hepatic function (albumin, PT, total bilirubin), renal function (urea, serum creatinine, KDIGO stage), severity scores (SAPS II and SOFA score), and hemodilution parameters. An effect of a covariate on a structural parameter was retained if it caused a decrease of Bayesian information criterion (BIC) and/or reduced the corresponding between subject variability (BSV) with *p* < 0.05. The objective function value reduction was tested for significance via a likelihood ratio test. Diagnostic graphics were used for evaluating the goodness-of-fit. Concentration profiles were simulated and compared with the observed data using a prediction corrected visual predictive check (PC-VPC) to validate the model. Empirical percentiles (percentiles of the observed data [5th, 50th and 95th], calculated either for each unique value of time, or pooled by adjacent time intervals) and theorical percentiles (percentiles of simulated data) were assessed graphically.

### 2.7. Dosing Regimen Evaluation

For dose simulation, a dummy data set was generated that included 10,000 subjects. Monte Carlo simulations were performed using the final model. Three dose regimens (mg/kg) were investigated. The optimal target chosen was the maximum concentration (at t = 1 h) defined in the range 60–80 mg/L [17]. Concentrations higher than 80 mg/L were defined as overdosing, and lower than 60 mg/L as underdosing. Time concentration profile curves obtained from simulated doses were plotted, and a visual inspection was performed.

## 3. Results

### 3.1. Patients

During the study period, 39 patients were included in the study: 15 in the Control group, and 24 in the V-A ECMO group. The demographic characteristics are presented in Table 1. The TBW of patients was 70 (65–84) kg and 75 (60–87) kg for control and ECMO groups, respectively. No significant difference was observed regarding the renal function according to the KDIGO classification; however, AKI was much more severe in ECMO vs. control patients, 18 (7–54) mL/min 120 (86–191) mL/min, respectively (*p* = 0.001).

Patients received 29 (24–33) mg/kg and 32 (30–35) mg/kg of amikacin in control and V-A ECMO groups, respectively.

### 3.2. Bacteriological Findings

Bacteriological data of patients are presented in Table 2. In both groups, lung was the main infectious source with a large predominance of ventilator-associated pneumonia. The most frequent bacteria in the control group was *Escherichia coli* (26%) while in the V-A ECMO group, the most implicated germs were *Klebsiella oxytoca* (25%) and *Pseudomonas aeruginosa* (25%).

### 3.3. Population Pharmacokinetic Model

A total of 215 plasma concentrations were recorded and used for the modeling process. The mean number of measurements per patient was 5.5. Individual amikacin curve profiles are presented on Figure 1. Data were best described by a two-compartment model with first-order elimination. The pharmacokinetic parameters of this model were clearance (CL), central volume of distribution (*V_1_*), intercompartmental clearance (Q), peripheral volume of distribution (*V_2_*). The selected residual variability was a proportional error model.

Inclusion of KDIGO on CL decreased the BIC by 107 units. The other significant covariates were TBW on V_1_, which reduced the BIC by 35 units and being supported with ECMO or not on V_2_, which reduced the BIC by 24 units. Implementing SOFA score, biological covariates, hemodilution parameters as covariate on CL, V_1_ and V_2_ did not improve the model. 

The final covariate model was:Cl = 4.45 × (−0.41) ^KDIGO = 1^ × (−0.59) ^KDIGO = 2^ × (−0.93) ^KDIGO = 3^
V_1_ = 8.77 × (TBW/72) ^0.014^
V_2_ = 15.90 × (0.79) ^ECMO^

The goodness-of-fit plots are depicted in the Appendix A. All parameters were well estimated with relative standard errors < 50% (Table 3). In the PC-VPC, the median of observed data was well included within the 90% confidence interval of simulated data (Figure 2). PC-VPC for each significant covariate are presented in Appendix A.

### 3.4. Dosing Regimen Simulations

The Bayesian estimates of the final model were used to describe amikacin pharmacokinetics obtained for each individual in the population, depending on the different chosen variables of interest (different stages of KDIGO, presence of the covariate ECMO or not and the TBW fixed at 70 kg). The Figure 3 shows the results in control and ECMO patients depending on the KDIGO stage (column). Cmax target is materialized by horizontal (60–80 mg/L) and vertical (t = 1 h) grey lines. First, the curves obtained for the ECMO group were upward translations of the curves obtained in the control group. Second, KDIGO stage 0 patients were significantly underdosed in both the control and ECMO groups, while KDIGO stage 3 patients were overdosed and even more so in the ECMO group. To achieve the pharmacokinetic target in all patients, dose adjustment simulations were performed. Thus, in KDIGO stage 0 group, increasing dose to 35 mg/kg and 40 mg/kg were simulated (Figure 4A). A 35 mg/kg dose of amikacin brought the Cmax down to the lower end of the therapeutic range in KDIGO stage 0 patients, whether patients were supported by V-A ECMO or not. A 40 mg/kg dose of amikacin ensured that most of these patients were within the therapeutic range. For KDIGO stage 3 patients, a 25 mg/kg dose administration prevented an amikacin overdosing (Figure 4B).

## 4. Discussion

Our study described a pharmacokinetic population’s model of amikacin in critically ill patients supported or not by V-A ECMO. Inter-individual variabilities were observed on CL, V_1_ and V_2_, which could be explained by three significant covariates: the effects of KDIGO on amikacin clearance, TBW on V_1_ and V-A ECMO support on V_2_.

Monocompartmental model is the best model for studies based on maximal and trough concentrations while bicompartmental model is the best for studies using entire pharmacokinetics of amikacin [18]. Our study used 5.5 samples per patient and permitted a better estimation of the volume of distribution (Vd). Estimated data of amikacin CL and Vd were 4.45 L/h, 8.77 L for V_1_ and 15.9 L for V_2_, respectively; these data were similar to those described in patients treated with ECMO, with amikacin CL and Vd 4.4 (3.4–5.2) L/h, 19.8 (16.8–20) L, respectively [14,19,20,21,22,23,24].

Renal function was the first covariate which had a major impact on pharmacokinetic of amikacin in ICU patients undergoing V-A ECMO. Indeed, hydrophilic antibiotics such as amikacin, have a major distribution in extracellular fluids and a total renal elimination, which explains why kidney function is so significant. Several authors studied the impact of creatinine clearance on amikacin pharmacokinetics [18,20]. This is why this biological covariate has regularly been used into models of amikacin pharmacokinetics [19,20,21,22,24,25]. However, creatinine clearance was mostly calculated using Cockroft and Gault formula which is not the recommended method to estimate renal function in critically ill patients. Conversely, we calculated creatinine clearance through UV/P formula. In our study, measured creatinine clearance was lower in the control group than in the ECMO group. We used the KDIGO classification based on plasmatic creatinine and urine output [15]; this parameter is well correlated with the severity of acute renal failure in terms of morbidity and mortality. This scale was chosen according to international recommendations [16]. Surprisingly, creatinine clearance and urine output studied separately were not significant in our model. In ICU patients, we demonstrated that the stage of acute kidney injury described by the KDIGO classification had a major impact on the adequate amikacin dosing. Indeed, our simulations showed that a 40 mg/kg amikacin dose was necessary in KDIGO stage 0 patients to obtain maximal plasma concentrations in the 60-80 mg/L range in control and V-A ECMO patients. However, a 25 mg/kg amikacin dose was sufficient in KDIGO stage 3 patients in both groups.

The second significative covariate was the TBW. In our population, 17% and 27% patients had a BMI > 30 kg/m^2^ in ECMO and control groups, respectively. Physiological changes in obese patients significantly influence drugs pharmacokinetics, such as distribution, protein binding and renal elimination. It has been estimated that, in obese patients, 40% of the aminoglycoside dose was distributed in adipose tissue [26]. The use of TBW induced an overestimation of amikacin distribution volume, with a risk of overdose. Although body surface area is rarely used as a covariate, Boidin et al. suggested that its use could decrease the risk of overdose in obese patients [14]. This parameter integrating both weight and height, could be less subject to variation in critically ill patients [27].

The third significative covariate was V-A ECMO support. To our knowledge, this is the first study to evaluate the effect of V-A ECMO in a population pharmacokinetics approach. In patients on V-A ECMO, highly significant changes in drug pharmacokinetics can occur by interactions with the ECMO support, drug characteristics, and patient characteristics. The ECMO circuit and membrane itself may function as an additional pharmacokinetics compartment by sequestering drugs, increasing Vd and changing drug clearance and elimination [8,28,29]. Although amikacin has been used for many years to treat severe infections in critically ill patients, only one case-control study compared an ECMO group with matched critically ill patients not supported by ECMO. The results indicated that there was no significant difference in amikacin peak and trough concentrations between both groups [12]. Few studies described amikacin monitoring in patients supported with ECMO: maximal concentrations were often below the recommended concentrations without further adaptation of its dosing. Our study shows an impact of the ECMO covariate on the pharmacokinetics of amikacin.

From an efficacity point of view, around a fifth of patients had underdosing/underdosage (<60 mg/L). The physiopathology of sepsis can explain the modifications of amikacin pharmacokinetics: increasing volume of distribution and/or hyper-clearance. In the two study groups of our population, 15% of patients had a hyper-clearance. In this case, we recommend increasing the initial amikacin dose. Conversely, 40% and 71% of patients had maximal concentrations above 80 mg/L in control and V-A ECMO groups, respectively with administered doses of 29 (24–33) and 32 (30–35) mg/kg. This is likely explained by the severity of AKI in ECMO group (18 mL/min vs. 120 mL/min in control, *p* < 0.05), inducing higher plasma concentrations. Moreover, only 4% of ECMO patients had a trough concentration lower than 2.5 mg/L (vs. 20% in control group). In septic shock, a high dose of amikacin is recommended without considering the renal function. This can explain the higher concentrations observed in this group. Finally, only 8% of ECMO patients had maximal concentrations in the recommended 60–80 mg/L range, underlining the need for a tailored pharmacokinetic approach. 

In the case of amikacin, only a few pharmacokinetic population’s studies have been carried out in ICU patients on ECMO, which does not allow us to compare our results. So, our simulations will require prospective clinical validation. 

Our study presents some limitations. First, the small size of the study groups limits the ability to highlight relevant covariates. However, we did not include patients treated with renal replacement therapy, a criterion that reduced our cohort. Second, we did not perform any external validation of our model. External validation is the most robust approach for model testing. It consists in applying the final model to a new population and determining its accuracy, reproducibility, and conditions of application; however, it was difficult to collect enough data from patients identical to the population of this study. Third, the time elapsed between ECMO implantation and administration of amikacin was not considered. However, there is probably a large intra-individual patient variability over time that can affect the distribution volume of amikacin. Inter-occasion variability was not integrated into our model. Similarly, the evolution is neither linear nor fixed throughout ICU stay. Fourth, we did not specifically focus on patients with creatinine hyper-clearance, which has a major impact on the clearance of drugs exclusively eliminated by kidneys. Finally, clinical success and microbiological response were not integrated in our model.

## 5. Conclusions

In this study, we describe a pharmacokinetic model of amikacin administered in critically ill patients on V-A ECMO. This population approach highlights three significant covariates which can explain interindividual variabilities of amikacin: KDIGO classification on its clearance, TBW on volume of the first compartment and ECMO on volume of the second compartment. Our simulations showed that the best dosage of amikacin was 40 mg/kg for KDIGO stage 0 patients while 25 mg/kg for KDIGO stage 3 patients was relevant. ECMO only had a secondary impact on amikacin pharmacokinetics as compared to acute kidney injury.

## Figures and Tables

**Figure 1 pharmaceutics-14-00289-f001:**
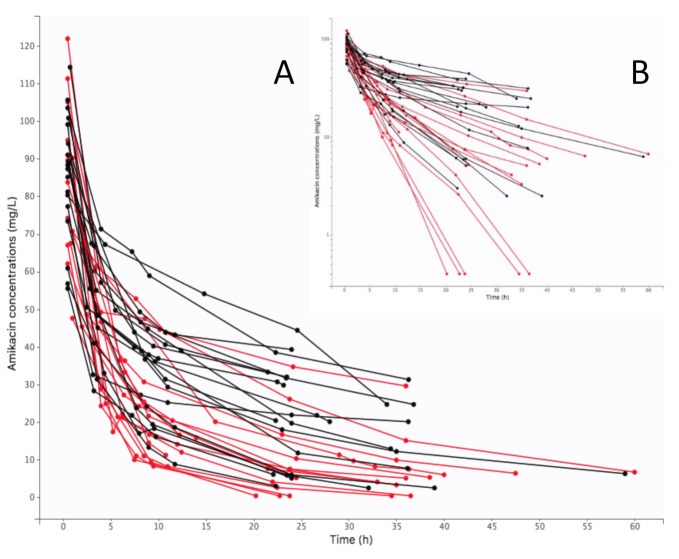
Individual concentration-time curves of amikacin in plasma (black color for V-A ECMO, red color for control): (**A**) arithmetic scale, (**B**) logarithmic scale.

**Figure 2 pharmaceutics-14-00289-f002:**
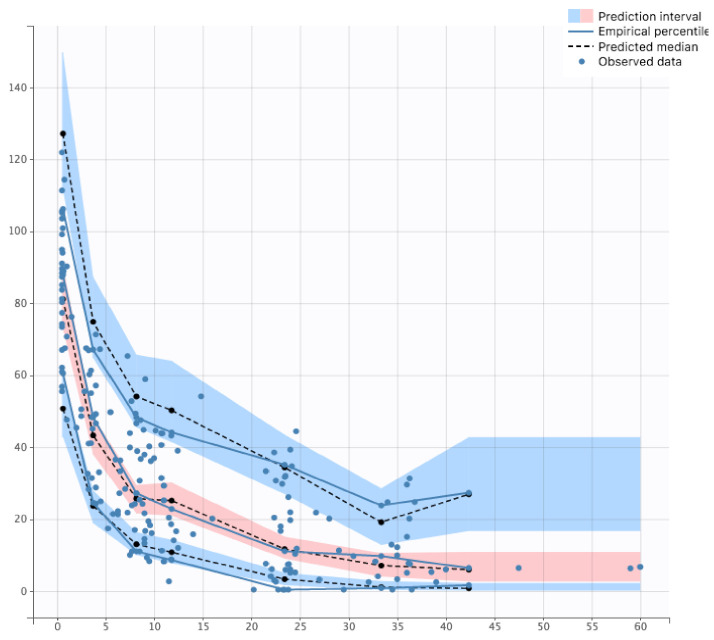
Prediction Corrected Visual Predictive Check of final model.

**Figure 3 pharmaceutics-14-00289-f003:**
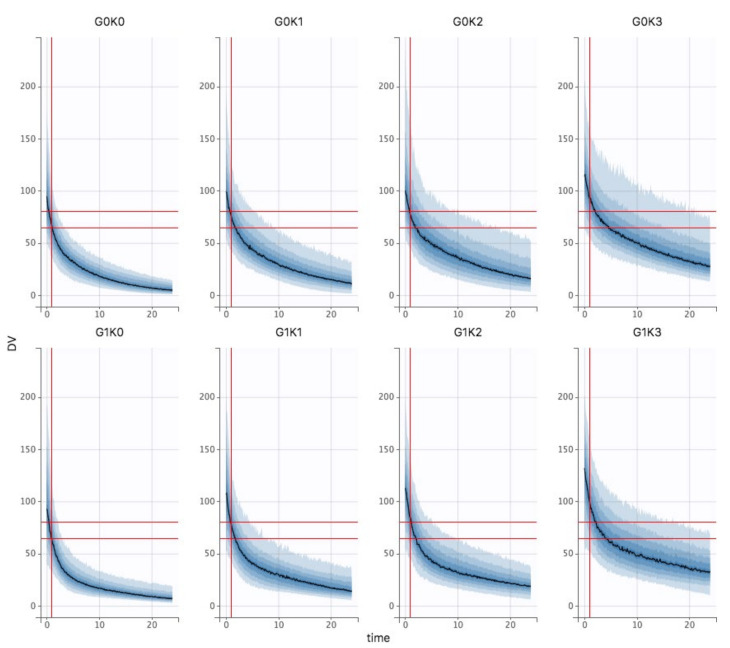
Bayesian estimates of the final model in control (top row) and V-A ECMO (bottom row) patients as a function of KDIGO stage (column).

**Figure 4 pharmaceutics-14-00289-f004:**
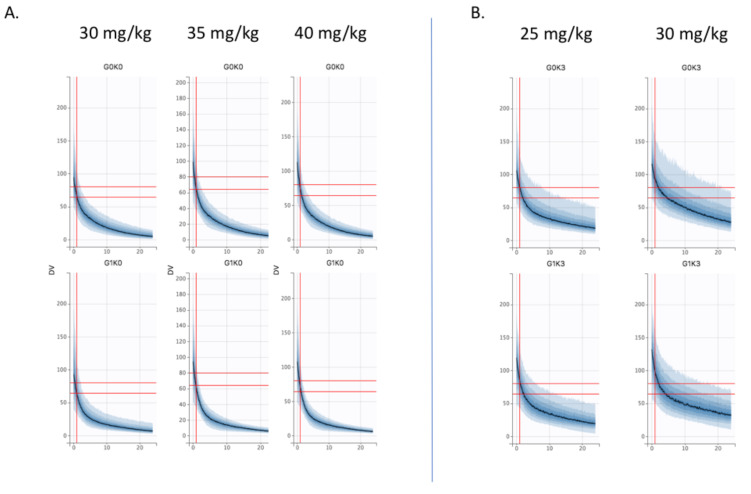
Dose simulations (**A**) at 35 mg/kg and 40 mg/kg in KDIGO stage 0 control (G0K0, top) and V-A ECMO (G1K0, bottom) patients vs. 30 mg/kg (**B**) at 25 mg/kg in KDIGO stage 3 control (G0K0, top) and V-A ECMO (G3K0, bottom) patients vs. 30 mg/kg.

**Table 1 pharmaceutics-14-00289-t001:** Demographic characteristics of the population.

	Control (*n* = 15)	V-A ECMO (*n* = 24)	*p*
Age (years)	62 (52–75)	60 (51–64)	0.23
Male sex	10 (67%)	16 (64%)	0.79
SAPS II	46 (39–58)	60 (52–69)	**0.01**
**Inclusion**			
Delay between ICU admission and amikacin injection (days)	5 (2–8)	5 (1–9)	0.47
SOFA	11 (9–12)	13 (11–16)	**0.01**
Total Body Weight (kg)	70 (65–84)	75 (60–87)	0.77
Height (cm)	168 (164–172)	173 (166–176)	0.11
BMI (kg/m^2^)	26 (24–30)	25 (22–29)	0.42
Norepinepinephrine dosing (μg/kg/min)	0.4 (0.1–1.3)	0.4 (0.1–1.5)	0.69
**Reason for admission**			
Postoperative/cardiac surgery	13 (87%)	11 (46%)	
Valve replacement	3 (23%)	5 (45%)	
Heart transplantation		3 (27%)	
Aorta replacement	2 (15%)	0	
Coronary artery bypass graft	4 (31%)	1 (9%)	
Other	4 (31%)	2 (18%)	
Medical	2 (13%)	13 (54%)	
Cardiogenic shock	1 (50%)	10 (77%)	
Dilated heart disease	1 (100%)	4 (40%)	
Ischemic heart disease	-	2 (20%)	
Myocarditis	-	1 (10%)	
Other	1 (50%)	3 (30%)	
Refractory cardiac arrest	-	3 (23%)	
**Reason of V-A ECMO support**			
Cardiogenic shock		13 (54%)	
Right heart failure		6 (25%)	
Refractory cardiac arrest		3 (13%)	
Graft dysfunction		2 (8%)	
**V-A ECMO characteristics**			
V-A ECMO		24 (100%)	
Flow rate (L/min)		4.2 (4–4.8)	
Flow rate indexed to weight (L/min/kg)		0.06 (0.05–0.07)	
**Biological results**			
Aspartate aminotransferase (mmol/L)	73 (37–98)	83 (48–138)	0.96
Alanine aminotransferase (mmol/L)	40 (28–112)	38 (26–106)	0.89
Total bilirubin (mmol/L)	13 (8–23)	37 (16–125)	0.01
Free bilirubin (mmol/L)	10 (6–18)	29 (10–97)	0.01
Conjugated bilirubin (μmol/L)	3 (2–5)	10 (4–23)	0.0006
Prothrombin time (%)	72 (55–77)	63 (54–69)	0.19
Protidemia (g/L)	55 (49–61)	48 (41–51)	0.02
Albumine (g/L)	19 (16–23)	20 (16–22)	0.998
Arterial lactate (mmol/L)	1.9 (1–2.5)	2 (1.6–2.3)	0.90
**Hemodilution parameters**			
24 h input/output balance (mL)	875 (438–1375)	1000 (500–1750)	0.95
**Kidney function**			
Creatinine clearance (mL/min)	120 (86–191)	18 (7–54)	0.001
KDIGO stage 0	10 (67%)	11 (44%)	0.20
KDIGO stage 1	2 (13%)	3 (12%)	0.94
KDIGO stage 2	2 (13%)	6 (24%)	0.77
KDIGO stage 3	1 (7%)	5 (20%)	0.83

**Table 2 pharmaceutics-14-00289-t002:** Bacteriologic findings.

	Control (*n* = 15)	ECMO (*n* = 24)
**Infectious site**		
Pneumonia	4 (27%)	14 (58%)
**- associated bacteremia**	-	5 (36%)
Pleuresy	1 (7%)	1 (4%)
Endocarditis	1 (7%)	-
Osteomyelitis	1 (7%)	-
Bacteriemia	-	-
Urinary tract infection	1 (7%)	-
Scarpa infection	-	2 (8%)
Catheter related	1 (7%)	3 (13%)
**- bloodstream infection**	1 (100%)	2 (67%)
No documentation	6 (40%)	4 (17%)
**Germs**		
*Acinetobacter baumanii*	-	4 (16%)
*Branhamella catarrhalis*	-	1 (4%)
*Citrobacter koseri*	-	1 (4%)
*Enterobacter aerogenes*	-	4 (16%)
*Enterobacter cloacae*	3 (20%)	3 (12%)
*Enterococcus faecalis*	2 (13%)	6 (25%)
*Escherichia coli*	4 (26%)	5 (20%)
**- extended spectrum beta-lactamase**	1 (6%)	-
*Haemophilus influenzae*	-	4 (16%)
*Hafnia alvei*	-	3 (12%)
*Klebsiella oxytoca*	-	6 (25%)
*Klebsiella pneumoniae*	2 (13%)	2 (8%)
*Proteus mirabillis*	1 (6%)	2 (8%)
*Pseudomonas aeruginosa*	-	6 (25%)
Methicillin-sensitive *Staphylococcus aureus*	1 (6%)	1 (4%)
*Veillonella* spp	-	1 (4%)

**Table 3 pharmaceutics-14-00289-t003:** Population pharmacokinetic parameters of amikacin from the final model.

Parameter	Estimate	RSE ^a^ (%)
CL (L/h)	4.45	6.33
V1 (L)	8.77	40.2
V2 (L)	15.90	20.0
Q (L/h)	6.96	17.5
KDIGO stage 1/CL	−0.41	35.4
KDIGO stage 2/CL	−0.59	18.8
KDIGO stage 3/CL	−0.93	14.9
TBW/V1	0.015	34.0
ECMO group/V2	0.76	35.8
ωCL *	0.25	13.0
ωV1 *	0.36	19.5
ωV2 *	0.63	19.1
Σprop * (%)	0.16	7.1

^a^ RSE: relative standard error (standard error of the estimate divided by the estimate and multiplied by 100); * ω, coefficient of variation for between-subject variability; σ, parameters of error model.

## Data Availability

Available on request to the corresponding author.

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
