# Peer review of "Population Pharmacokinetics of Amikacin in Patients on Veno-Arterial Extracorporeal Membrane Oxygenation"

_pharmaceutics, 2022, doi:10.3390/pharmaceutics14020289_

Round 1

Reviewer 1 Report

The topic of the manuscript is interesting and fits well the scope of Pharmaceutics. The reviewer feel it can be accepted after minor amendments.

1) The race should be indicated in Demographic characteristics of the population.

2) As the journal only has electronic version, there is no extra color page charge. The figures should be  colorful. It will greatly enhance the presentation. 

3) The authors should mention the elimination pathway of the drug as background information. 

Reviewer 2 Report

The manuscript titled “Population pharmacokinetics of amikacin in patients on veno-arterial extracorporeal membrane oxygenation” attempted to assess the population pharmacokinetics of amikacin.

As such the manuscript is well written. However, there are some deficiencies in the manuscript which need to be addressed to make it publishable:

  1. Were physiological/pathological conditions of the patients considered in building the models?
  2. Stratification of patient characteristics will be useful in analyzing the results.
  3. Authors need to include Parameter Sensitivity Analysis (PSA) data to validate the model.
  4. There wasn’t any description of the software in the manuscript and needs to be included.
  5. Data used is >6-year-old. Can recent data be added in the study?
  6. AST and ALT are more commonly written like this rather than ASAT and ALAT.
  7. Tables should be in text format rather than image. Some of the tables are not clear due to this.

Reviewer 3 Report

The manuscript describes population pharmacokinetics of amikacin in patients on veno-arterial extracorporeal membrane oxygenation. The study is well planed and performed, taking into account that always is difficult to form a group of patients with the same diagnosis. The manuscript is clear and well organized. The methods are suitable and the population model was properly selected. The manuscript can be published after minor revision. The following points can be improved:

The time of the start of the treatment of every patient can be given in a table as a supplementary material.

Missing part is description of analysis of the concentrations in plasma. How sensitive was the applied method. Please, explain.

The headings of Figure 4. Is not very clear, if it is placed in 2 pages. It will be better to be presented in 1 page.

Line 251 through the whole manuscript: Please change “plasmatic concentrations” to “plasma concentrations”

It could be useful if the authors give data for inter-individual variability. May be it will be nice for readers if the calculated parameters are given in the supplementary file.
